# Costatone C—A New Halogenated Monoterpene from the New Zealand Red Alga *Plocamium angustum*

**DOI:** 10.3390/md17070418

**Published:** 2019-07-19

**Authors:** Joe Bracegirdle, Zaineb Sohail, Michael J. Fairhurst, Monica L. Gerth, Giuseppe C. Zuccarello, Muhammad Ali Hashmi, Robert A. Keyzers

**Affiliations:** 1School of Chemical and Physical Sciences, Victoria University of Wellington, Wellington 6012, New Zealand; 2Centre for Biodiscovery, Victoria University of Wellington, Wellington 6012, New Zealand; 3Maurice Wilkins Centre for Molecular Biodiscovery, Wellington 6012, New Zealand; 4Department of Life Sciences, University of Management and Technology, C-II, Johar Town Lahore 54770, Pakistan; 5School of Biological Sciences, Victoria University of Wellington, Wellington 6012, New Zealand; 6Department of Chemistry, University of Education, Attock Campus, Attock 43600, Pakistan

**Keywords:** halogenated monoterpene, *Plocamium angustum*, costatone, Mosher’s analysis, DFT, ECD

## Abstract

Red algae of the genus *Plocamium* have been a rich source of halogenated monoterpenes. Herein, a new cyclic monoterpene, costatone C (**7**), was isolated from the extract of *P. angustum* collected in New Zealand, along with the previously reported (1*E*,5*Z*)-1,6-dichloro-2-methylhepta-1,5-dien-3-ol (**8**). Elucidation of the planar structure of **7** was achieved through conventional NMR and (−)-HR-APCI-MS techniques, and the absolute configuration by comparison of experimental and DFT-calculated ECD spectra. The absolute configuration of **8** was determined using Mosher’s method. Compound **7** showed mild antibacterial activity against *Staphylococcus aureus* and *S. epidermidis*. The state of *Plocamium* taxonomy and its implications upon natural product distributions, especially across samples from specimens collected in different countries, is also discussed.

## 1. Introduction

Red algae, found in marine habitats around the world, generate a huge range of structurally diverse natural products. New Zealand is home to a large variety of red algal species, including the genus *Plocamium.* This genus has been a prolific source of halogenated monoterpenes, often showing antibacterial or cytotoxic bioactivity [1], and is therefore of interest for drug discovery. 

*P. angustum* is found throughout the Pacific Islands and the length of New Zealand, and is also very common around Australia, the origin of the alga for all previous published chemical investigations. In 1979, Dunlop et al. reported the isolation of the novel bromodichloro monoterpene **1** alongside the previously reported dienone **2** from an alga collected from South Australia [2]. Two separate studies focusing on algae collected in Victoria, Australia, resulted in the isolation of three other highly halogenated monoterpenes (**3**–**5**) (Scheme 1), where plocamenone (**4**) showed both antibacterial and antifungal activity [3,4]. Thus far, there have been no reports of studies on *P. angustum* collected in New Zealand.

Recent studies have shown that morphological species identification in algae is not accurate in determining species status, i.e. independent evolutionary lineages, suggesting that only with molecular data can morphologically similar but distinct species be determined [5]. Incorrect species identification could explain some of the differences in natural product content, if such contents are species specific. *Plocamium* is a genus to which molecular taxonomic methods have been applied, revealing that morphological identification, especially of the commonly named entity *P. cartilagineum*, does not conform to evolutionary lineages [6,7]. In New Zealand, six named species of *Plocamium* have been reported [8], while a recent study indicates that at least eleven species are found in New Zealand, and that there are no morphological characters to distinguish among these species [9].

Costatone A (**3**) [10] and B (**6**) [11] are cyclic polyhalogenated monoterpenes, first isolated from different *P. costatum* samples collected in South Australia. The absolute configuration of costatone A was solved by X-ray crystallography, and has since been isolated from samples identified as *P. angustum* [4]. As part of our investigation into the secondary metabolites of common Rhodophytes collected from the coast of Wellington, New Zealand [12], it was observed using ^1^H NMR spectroscopy that an extract of an alga identified as “*P. angustum”* had many chemical shifts indicative of halogenated monoterpenes. A detailed analysis of the alga resulted in the isolation of the two major compounds that have not been previously reported from “*P*. *angustum”*, new compound **7** and known compound **8** (Scheme 1). Costatone C (**7**) is the first polyhalogenated monoterpene with a tetrahydropyran ring isolated from this species. 

## 2. Results and Discussion

The methanolic extract of *P. angustum* (50.0 g wet weight) was partitioned over polystyrene(divinylbenzene) using increasing percentages of acetone in H_2_O. The ^1^H NMR spectrum of the 75% acetone in H_2_O fraction showed many resonances attributable to halogenated monoterpenes, and was consequently purified by size-exclusion chromatography, yielding two major components by TLC. Reversed-phase HPLC was then used to purify the resulting compounds.

Compound **7** was isolated as a yellow oil. A molecular formula of C_10_H_14_OCl_2_Br_2_ was established from negative ion high resolution atmospheric-pressure chemical ionisation mass spectrometry ((−)-HR-APCI-MS) analysis. This molecular formula is indicative of a monoterpene, and requires two degrees of unsaturation, one accounted for by an alkene as evidenced by the two downfield ^13^C resonances (in CD_3_OD as CDCl_3_ had overlapping resonances) *δ*_C_ 138.5 and 117.5 (Table 1). The ^13^C NMR and multiplicity-edited HSQC spectra also showed signals for a non-protonated carbon (*δ*_C_ 73.6), three methines (*δ*_C_ 83.4, 73.8, 54.5), two methylenes (*δ*_C_ 39.1, 29.0) and two methyl groups (*δ*_C_ 28.9, 13.1), with the olefinic carbon C-1 (*δ*_C_ 117.5) also bearing one hydrogen (H-1 *δ*_H_ 6.28). The resonance for H-1 appears as a quintet (*J =* 1.4 Hz) and shows a COSY correlation to the vinylic methyl, along with an allylic coupling to H-3 (*δ*_H_ 4.22). Further COSY correlations from H-3 assigned the remainder of the spin system, first to methylene H_2_-4 (*δ*_H_ 2.45, 2.15) and methine H-5 (*δ*_H_ 4.75). A second spin system deduced from COSY correlations connected the other oxygenated methine H-7 (*δ*_H_ 4.29) to methylene H_2_-8 (*δ*_H_ 3.94, 3.73). These two fragments are linked via non-protonated C-6 (*δ*_C_ 73.6), based upon two- and three-bond HMBC correlations from H_2_-4, H-5, H-7 and H_2_-8. As the molecular formula requires another degree of unsaturation, and H-3 correlates in the HMBC spectrum to C-7, the molecule must be cyclic, containing a tetrahydropyran moiety (Figure 1).

The ^13^C NMR signal for C-6 at *δ*_C_ 73.6 was split into an asymmetric doublet, with an intensity ratio of ~3:1, a phenomenon known as the chlorine isotope effect [13], thus there must be a chlorine present at this centre. The locations of the remaining halogens were then assigned on the basis of chemical shift arguments. Motti et al. isolated costatols C–E (**9**–**11**) from *P. costatum,* (Scheme 1), where each also have an *E* trisubstituted double bond, however **9** and **10** are chlorinated while **11** is brominated at position C-1 [11]. Clearly, the ^13^C NMR data of C-1 and C-2 for **7** (*δ*_C_ 117.5 and 138.5) align with a chlorovinyl group (**9**
*δ*_C_ 116.1 and 138.8, **10**
*δ*_C_ 116.2 and 138.7, **11**
*δ*_C_ 105.6 and 141.7), thus the other chlorine was assigned at C-1. As no other ^13^C signals were split, assigning the chlorine atom to a sp^2^ carbon is in agreement with the note that the splitting decreases with increased *s* character of the carbon atom [13]. This necessitates bromine substituents to be at C-5 and C-8, which was further evidenced by their more shielded ^13^C shifts (*δ*_C_ 54.5 and 29.0, respectively).

With the planar structure of **7** in hand, the geometry of the alkene and relative configuration of the four chiral centres were deduced from through space 2D ROESY NMR correlations and ^1^H NMR coupling constants. The double bond must have an *E* geometry on the basis of the through-space ROESY correlation between H-1 and H-3, with ^13^C NMR data consistent with that of co-isolated compound **8** that also possess a similar *E*-chloroalkene [14,15]. H-3 and H-5 both show ROESY correlations to the same proton H-4_b ­_(2.15 ppm), with smaller coupling constants (*J_H-3/H-4b_* = 2.6 Hz, *J_H-4b/H-5_* = 4.4 Hz) indicative of axial/equatorial relationships, whereas no ROESY correlations were observed to H-4_a_, but large coupling constants suggest axial/axial couplings (*J_H-3/H-4a_* = 11.7 Hz, *J_H-4a/H-5_* = 12.0 Hz), therefore H-3 and H-5 are *syn*, corroborated by their shared nOe correlation. The through-space ROESY correlations among H-5, H-8_a_ and methyl H_3_-10 put these substituents on the same side of the ring as well, further evidenced by H-8_a_’s correlation to H-3. Thus, the relative configuration *3R, 5R, 6S, 7R* was deduced as depicted (Scheme 1). This favoured conformation minimises 1,3-diaxial interactions of the chair by orientating most of the bulky groups in equatorial positions, consistent with the observed scalar ^1^H coupling constants.

Where crystallographic data are unobtainable, computational chemistry can play a significant role in establishing the absolute configuration of a compound if experimental electronic circular dichroism (ECD) data are available. Computation of ECD data and their comparison to experimental data can lead to the assignment of absolute configuration. For this purpose, the structure of **7** (Figure 2) was optimised at the PBE0-D3BJ/aug-cc-pVTZ/SMD_MeOH_ level of theory followed by a relaxed scan by varying two key dihedrals (C4-C3-C2-C9 and C6-C7-C8-Br) in 24 steps of 15° each. After that, the lowest energy conformations were selected from the resulting potential energy surface (PES) as shown (Appendix A). After removal of duplicates, ten conformers were subjected to ECD computations using time-dependent density functional theory (TDDFT) and the results were compared with the experimental CD spectrum obtained in MeOH after summing based upon their Boltzmann weightings (Figure 3). The computed ECD spectrum was scaled for its peak intensity and wavelength to match with the experimental spectrum [16,17,18]. The computed ECD spectrum is in a very good agreement to the experimental, which confirms the absolute configuration of compound **7** as 3*R*, 5*R*, 6*S*, 7*R*, as shown in Figure 3.

The dichlorinated bisnor-monoterpene (1*E*,5*Z*)-1,6-dichloro-2-methylhepta-1,5-dien-3-ol (**8**) was also isolated as the major metabolite. Terpene **8** was initially reported from *P. cruciferum* and was identified here by comparison to the reported NMR and EI-MS data [14,15]. The absolute configuration at C-3 was not determined originally, therefore Mosher’s ester analysis was used to derivatise the secondary alcohol [19]. Both *R*- and *S*-MTPA esters were produced under Steglich conditions [20], with subsequent NMR analysis leading to the conclusion that C-3 has an *S* configuration (Figure 4). As the observed optical rotation (−22) agrees with that previously reported (−9.8), this established the absolute stereostructure of the *P. cruciferum* metabolite [14,15].

The antibacterial properties of **7** and **8** were assessed using *Pseudomonas aeruginosa* (Gram negative), *Staphylococcus aureus* (Gram positive) and *Staphylococcus epidermidis* (Gram positive) (Appendix A). Although no inhibitory activity was detected against *P. aeruginosa*, **7** showed mild activity against both *S. aureus* and *S. epidermidis*, with minimum inhibitory concentrations (MIC) of 128 and 64 µM, respectively. No antibacterial activity was observed for **8**.

Although morphologically identified as *P. angustum*, phylogenetic analysis with the cytochrome oxidase subunit 1 gene (Figure 5) and the ribulose bisphosphate carboxylase large subunit (data not shown) confirmed that this alga is a cryptic species G [9]. Comparison with other available sequences indicates that this species is found in the Wellington region and along the Wairarapa coast (southeast North Island). This alga is distinct both from other New Zealand species and from Australian species identified as *P. angustum* (Appendix A) [9]. The sample is most similar to sequences identified as *P. cartilagineum* from New Zealand, however *P. cartilagineum* is a European species and therefore is unlikely to be a correct identification. This indicates that additional interesting chemistries could be discovered within the many cryptic species of *Plocamium* found in New Zealand and around the world.

## 3. Conclusions

Red algae of the genus *Plocamium* continue today to be a rich source of new halogenated monoterpenes. Through chromatographic techniques, the dibromo-dichloro-tetrahydropyran costatone C **7** was purified, and its structure and relative configuration solved by MS and NMR. By comparison of its theoretical and experimental ECD spectra, the absolute configuration was solved. As our study focused on an alga collected in New Zealand, there is clearly a different chemotype relative to those previously reported from Australia, and this is possibly due to the different species status of these morphologically similar algae, warranting further investigation into the metabolomics and taxonomy of these species.

## 4. Materials and Methods

### 4.1. General Procedures

Optical rotations were measured using a Rudolph Autopol II polarimeter. ECD spectra were recorded on a ChiraScan CD spectrometer (Applied Photophysics, Surrey, United Kingdom). A 600 MHz Varian Direct Drive spectrometer equipped with a 5 mm PFG dual broadband probe was used to record the NMR spectra of **7**, **8** and **8a**,**b** (600 MHz for ^1^H nuclei and 150 MHz for ^13^C nuclei). The residual solvent peak was used as an internal reference for ^1^H (*δ*_H_ 3.31, CD_3_OD; 7.26, CDCl_3_) and ^13^C (*δ*_C_ 49.0, CD_3_OD; 7.16, CDCl_3_) chemical shifts [21]. High-resolution (APCI) mass spectrometric data were obtained with an Agilent 6530 Accurate Mass Q-TOF LC-MS (Santa Clara, CA, USA) equipped with a 1260 Infinity binary pump. IR (film) spectra were recorded using a Bruker Platinum Alpha FTIR spectrometer (Leipzig, Germany). EI mass spectrometric data were acquired using a Shimadzu 2010 Plus gas chromatograph (Kyoto, Japan) operating with a GCMS-QP2010 MS detector.

Reversed-phase column chromatography was achieved using Supelco Diaion HP20 (PSDVB) chromatographic resin. Size exclusion chromatography was achieved using Sephadex LH20 resin. HPLC purifications were carried out using either an Agilent Technologies 1260 Infinity HPLC equipped with a diode array detector or an Agilent 380 evaporative light-scattering detector (ELSD), using an octadecyl-derivatised silica (C_18_, 5 µm, 100 Å) HPLC column (Phenomenex; 4.6 mm × 250 mm, flow rate: 1 mL/min). All solvents used for column chromatography were of HPLC grade and H_2_O was glass distilled. Solvent mixtures are reported as per cent v/v unless otherwise stated. TLC was carried out using Machery-Nagel Polygram Sil G/UV_254_ plates, run in 1:3 EtOAc:hexanes and developed using a H_2_SO_4_ (5% in MeOH)/vanillin (0.1% w/v in EtOH) char. 

### 4.2. Collection of *Plocamium angustum*

Specimens of *Plocamium angustum* were collected by hand using scuba at a depth of 3−10 m from Moa Point, Wellington, New Zealand, in January 2017 and stored at −20 °C until extraction. A voucher specimen (JB06_38) is held at the School of Chemical and Physical Sciences, Victoria University of Wellington, New Zealand.

### 4.3. Extraction and Isolation

Frozen *P. angustum* (50.0 g wet weight) was extracted in MeOH (200 mL) twice overnight. The second extract, followed by the first, were passed through a HP20 column (30 mL), pre-equilibrated in H_2_O and combined following elution. The eluent was then diluted with an equal volume of water and passed back through the column twice, followed by a 100 mL H_2_O wash. The column was then eluted with 100 mL portions of: (1) 30% Me_2_CO/H_2_O; (2) 75% Me_2_CO/H_2_O; and (3) Me_2_CO (fractions A1−A3, respectively). Fraction A2 (300 mg) was then partitioned on Sephadex LH20 with 100% MeOH, and the resulting fractions were pooled together on the basis of TLC, resulting in two major fractions B1 (test tubes 36–43) and B2 (test tubes 44–51). A portion of sample B1 (20 mg) was subjected to silica gel chromatography (5:1 EtOAc:hexanes) to afford **8** (6.8 mg) as a pale yellow oil. A portion of sample B2 (10 mg) was further purified on a semipreparative C18 HPLC column (80% MeOH/H_2_O), yielding compound **7** (4.8 mg, t_R_ = 13.0 min) as a colourless oil. 

*Costatone C* (**7**): pale yellow oil, ${\left[\alpha \right]}_{D}^{20}$ −7.2 (*c* 0.05, CHCl_3_); IR υ (thin film): 2930, 1642, 1434, 1382, 1102, 1080, 782, 738, 587, 519 cm^-1^; ^1^H and ^13^C NMR data, see Table 1; (-)HR-APCI-MS m/z 376.8719 [M − H]^−^ (calcd. for C_10_H_13_OCl_2_Br_2_, 376.8716).

*Compound **8***: colourless oil, ${\left[\alpha \right]}_{D}^{20}$ −22 (*c* 0.1, CHCl_3_); NMR and MS data consistent with published [14,15].

### 4.4. Preparation of MTPA esters 12a and 12b

A solution of EDC.HCl (12 mg, 64 µmol), *S*-(+)-α-methoxyphenylacetic acid (15 mg, 64 µmol), and DMAP (14.8 mg, 120 µmol) was stirred in dry CH_2_Cl_2_ (0.5 mL) under Ar at 0 °C for 10 min after which **8** (4 mg, 12 µmol) in CH_2_Cl_2_ (0.5 mL) was added. The solution was allowed to come to room temperature and stirred under Ar for 48 h. CH_2_Cl_2_ (10 mL) was added and the mixture was washed in turn with 10% HCl (10 mL), H_2_O (10 mL), sat. NaHCO_3_ (10 mL) and H_2_O (10 mL) before being dried under reduced pressure. The sample was purified by flash silica gel chromatography (10:1 hexanes:EtOAc) to yield the crude product **12a**, analysed without further purification. The procedure was repeated with *R*-(-)-α-methoxyphenylacetic acid to yield product **12b** (see Appendix A).

*Compound **12a**:*^1^H NMR (CDCl_3_, 600 MHz) *δ*_H_ 7.48–7.40 (5H, m, aromatics); 6.25 (1H, s, H-1); 5.50 (1H, dd, H-3); 5.18 (1H, t, H-5); 3.51 (3H, s, OMe); 2.56 (2H, m, H-4); 2.03 (3H, s, H-7); 1.75 (3H, d, H-8).

*Compound **12b**:*^1^H NMR (CDCl_3_, 600 MHz) *δ*_H_ 7.48–7.40 (5H, m, aromatics); 6.16 (1H, s, H-1); 5.48 (1H, dd, H-3); 5.32 (1H, t, H-5); 3.55 (3H, s, OMe); 2.60 (2H, m, H-4); 2.09 (3H, s, H-7); 1.64 (3H, s, H-8).

### 4.5. Computational Data

All computations were performed using Gaussian 09 (Revision D.01) [22]. Density functional theory (DFT) was used for all the calculations utilising Adamo’s hybrid [23] version of Perdew, Burke and Ernzerhof functional (PBE0) [24,25] along with the application of Grimme’s empirical dispersion correction (D3) with Becke–Johnston damping (D3BJ) [26,27,28]. All calculations were performed with Ahlrich’s triplet ζ basis set def2-TZVP [29] supported by the Polarisable Continuum Model (PCM) with the integral equation formalism variant (IEFPCM) [30,31,32,33,34,35,36] for solvation modelling. The solvent for optimisation and ECD calculation was MeOH which was modelled with the SMD parameter set by Cramer and Truhlar [37] (as implemented in Gaussian 09) [22]. Calculated ECD spectra were scaled for both intensity and frequency to the experimental data (Appendix A). Frequency calculations at the same level of theory were used to confirm all the optimised structures to be true minima on the potential energy surface with the absence of imaginary frequencies. The 3D images of optimised molecules were drawn using CYLview [38] program.

### 4.6. Antibacterial Bioassay

*Pseudomonas aeruginosa* (PAO1) or *Staphylococcus aureus* (ATCC 25923) were used to inoculate 100 µL of Mueller Hinton broth (Formedium; Hunstanton, UK) amended with 100 µg/mL of the test compounds in a 96-well plate (control wells contained an equivalent volume of DMSO). Cells were incubated at 37 °C, shaking at 600 RPM, for 24 h (Incumix, Select Bioproducts; Edison, NJ, USA). The optical density was measured at 600 nm (Enspire 2300 Multilabel Reader, Perkin Elmer; Waltham, MA, USA) and the absorbance value of the media-only controls were averaged and subtracted from all measurements. Values were calculated from three replicates.

*S. aureus* and *S. epidermidis* (ATCC 35984) were then tested with **7** to determine the strength of inhibition in Gram-positive bacteria. Similar to the previous experiment, *S. aureus* and *S. epidermidis* were used to inoculate 100 µL of Mueller Hinton broth, amended with a 2-fold dilution series of **7** from 0.5 µg/mL to 128 µg/mL in a 96-well plate (control wells contained an equivalent volume of DMSO). Cells were incubated at 37 °C, shaking at 600 RPM, for 24 h. The optical density was measured at 600 nm and the absorbance value of the media-only controls were averaged and subtracted from all measurements. Values were calculated from three replicates.

### 4.7. Molecular Analysis

DNA extraction, PCR amplification, and sequencing of the cytochrome oxidase genes followed previously described method [9]. Various sequences of *Plocamium* were downloaded from Genbank or were gained directly from [9]. Phylogenetic trees were made using RAxML 8 [39] to construct maximum-likelihood trees (ML) to show the most likely tree from the dataset. RAxML was performed using the GTR+gamma model. The reliability of the ML topologies was evaluated based on 1000 nonparametric bootstrap replicates [40].

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
