# Peer review of "Costatone C—A New Halogenated Monoterpene from the New Zealand Red Alga Plocamium angustum"

_marinedrugs, 2019, doi:10.3390/md17070418_

Reviewer 1 Report

Marine Drugs -June 2019

Review Report

Title: Costatone C – a New Halogenated Monoterpene from the New Zealand Red Alga “Plocamium angustum”

The paper describes the isolation and characterisation of a new halogenated monoterpene including relative and absolute configurations. It also showed mild anti-microbial activity. Importantly through molecular genome analysis the paper has helped cleared the taxonomic classification of this genus which has often been problematic though morphological analysis only as done previously.

Broad comments:

The paper is well written apart from the three schemes used for structural drawings. It is recommended that schemes 2-3 should be combined with scheme 1.

Specific comments

1.       Line 62: compounds 7 and 8….replacing compound:

2.       Line 63:  delete the and replace with this. Should read: ….from this species.

3.       Lines 68-75: Compound numbering in scheme 1 should follow a logical order either going from left to right or up, down.

4.       Lines 115-116:  ……..add the words: ‘at position C-1’ to read: …while 11 is brominated at position C-1 [11].

5.       Lines 123-124 (Table 1): At multiplicity of carbons in the 13C column (to be consistent with Marine Drugs format).

6.       Line 128: add compound 8 to read: …co-isolated compound 8….

7.       Line 129: add the word ‘possess’ to read…..also possess a similar….

8.       Line 133: replace nOe with rOe

9.       Line 197:delete ‘in’ and replace with ‘a’ to read ..alga is a cryptic…..

10.   Line 317:add the words: previously described method to read…….genes followed previously described method [9].

Author Response

On behalf of my co-authors and I, we would like to thank you and the anonymous reviewers who have evaluated our manuscript. We would like to particularly acknowledge the very rapid and efficient review process for this manuscript. We were also extremely gratified at the reviewer’s comments regarding the high quality of the writing and the science of our manuscript.

The paper is well written apart from the three schemes used for structural drawings. It is recommended that schemes 2-3 should be combined with scheme 1. We have done this.

Specific comments

1.        Line 62: compounds 7 and 8….replacing compound: Done.

2.        Line 63:  delete the and replace with this. Should read: ….from this species. Done.

3.       Lines 68-75: Compound numbering in scheme 1 should follow a logical order either going from left to right or up, down. Done.

4.       Lines 115-116:  ……..add the words: ‘at position C-1’ to read: …while 11 is brominated at position C-1 [11]. Done.

5.       Lines 123-124 (Table 1): At multiplicity of carbons in the 13C column (to be consistent with Marine Drugs format). Done.

6.       Line 128: add compound 8 to read: …co-isolated compound 8…. Done.

7.       Line 129: add the word ‘possess’ to read…..also possess a similar…. Done.

8.       Line 133: replace nOe with rOe. We believe that while the through-space excitation is detected using a ROESY experiment, the phenomenon observed is still a nuclear Overhauser effect so we have retained the nOe description.

9.       Line 197:delete ‘in’ and replace with ‘a’ to read ..alga is a cryptic….. Done.

10.   Line 317:add the words: previously described method to read…….genes followedpreviously described method [9]. Done.

Reviewer 2 Report

The calculated CD spectrum of compound 7 in figure 3 is not in very good agreement to the experimental one, it is in an unreliable agreement with experiment. Is it possible that both spectra have the same identical frequency position, the same bandwidth, and even the same small spikes around the maxima? In tens of similar calculations I have made on natural products, I have never met something similar.

So, before doubting of the authors’ good faith, I would like to know something more about the calculations.

1. Have you performed a true conformational analysis of 7? I guess that the geometry depicted in Fig. 2 is the absolute minimum, but is it possible that there are no relative minima at higher energy? And how much distanced in energy from the absolute one?

2. What about the UV spectrum? In such cases you must first do a calculation on the electronic transitions of the molecule, and only then switch to the Cotton Effect of each transition.

This information is needed at least as supplementary material. That’s the way researchers in the theoretical field work, when making comparisons between experiment and theory.  

Apart from this major drawback, there are other things I would like to note, as that phylogenetic analysis is not very appropriate for a chemistry journal as Marine drugs, so that fig. 5 should be eliminated. It’s enough to say that recent phylogenetic analysis demonstrates that the taxonomic status of former Plocamium angustum is still to be well assessed. This is happening for thousands of living beings, in this case you keep a voucher specimen and then refer to it.

Also the whole manuscript is somehow redundant in language, in particular the discussion of the NMR spectra of 7 should be reduced.

Author Response

On behalf of my co-authors and I, we would like to thank you and the anonymous reviewers who have evaluated our manuscript. We would like to particularly acknowledge the very rapid and efficient review process for this manuscript. We were also extremely gratified at the reviewer’s comments regarding the high quality of the writing and the science of our manuscript.

The calculated CD spectrum of compound 7 in figure 3 is not in very good agreement to the experimental one, it is in an unreliable agreement with experiment. Is it possible that both spectra have the same identical frequency position, the same bandwidth, and even the same small spikes around the maxima? In tens of similar calculations I have made on natural products, I have never met something similar. The reviewer is rightfully concerned about the agreement. The calculated ECD spectrum was scaled for the peak intensity as well as for frequency and the final scaled version has been presented in the manuscript. To clarify the situation, we have presented a series of spectra below where a comparison of the experimental spectrum with the calculated unscaled, calculated with only intensity scaling, and calculated with both intensity and frequency scaling (present in the manuscript) have been made. Due to much lower intensity of the calculated values, the values were multiplied by 20. The frequency was also shifted 7 nm to the left for the calculated one to compare perfectly to the experimental spectrum whose data was available for 190-280 nm. We have included this figure in the supplementary information file (figure S23).

1. Have you performed a true conformational analysis of 7? I guess that the geometry depicted in Fig. 2 is the absolute minimum, but is it possible that there are no relative minima at higher energy? And how much distanced in energy from the absolute one? A conformational analysis was not performed for the compound under study. Instead, it was modelled in GaussView followed by a preliminary optimization on a smaller level of theory. Then the resultant structure was submitted to the level of theory described in the manuscript.

2. What about the UV spectrum? In such cases you must first do a calculation on the electronic transitions of the molecule, and only then switch to the Cotton Effect of each transition. ECD calculations were done using TD-DFT as a single point UV calculation as implemented in Gaussian. This gives both the UV and ECD spectra.

This information is needed at least as supplementary material. That’s the way researchers in the theoretical field work, when making comparisons between experiment and theory.  The above comparison of scaled and unscaled spectra has been provided as supporting information.

Apart from this major drawback, there are other things I would like to note, as that phylogenetic analysis is not very appropriate for a chemistry journal as Marine drugs, so that fig. 5 should be eliminated. It’s enough to say that recent phylogenetic analysis demonstrates that the taxonomic status of former Plocamium angustum is still to be well assessed. This is happening for thousands of living beings, in this case you keep a voucher specimen and then refer to it. We respectfully disagree with the reviewer regarding this point. The readership of Marine Drugs is wider than just chemists, who we agree may not find this phylogenetic tree useful, but we believe taxonomists and chemical ecologists in particular should find merit in its inclusion. Moreover, this figure gives relevance to our point that the taxonomy of Plocamium spp. across international borders is uncertain and more work on the relationship between taxonomic identification and genetic identity is required.

Also the whole manuscript is somehow redundant in language, in particular the discussion of the NMR spectra of 7 should be reduced. We are unsure what the reviewer would like us to do here, and we believe our rigorous assignment of the structure of our compound is far more important than not providing enough detail as is trend at the moment. We have left the text untouched in this regard.

Reviewer 3 Report

The manuscript describes the chemical investigation of a red alga Plocamium angustum, which yielded a new cyclic monoterpene (7). The structures of the isolated compounds were determined via NMR and MS analysis. The manuscript was well written and the structure elucidation was properly done. ECD DFT calculation (for 7) and Mosher’s method (for 8) were performed to determine the absolute configurations. The structure of the new compound 7 is interesting and will be interest of chemists. Furthermore, a mild antibacterial activity of compound 7 was discovered. The manuscript is recommended after minor revisions. A few queries and typographical errors:

1. The author should choose between “algae” or “alga”

2. Page 7 line 262, should be C10H13OCl2Br2

3. Instrument details for optical rotation measurement should be included

Author Response

On behalf of my co-authors and I, we would like to thank you and the anonymous reviewers who have evaluated our manuscript. We would like to particularly acknowledge the very rapid and efficient review process for this manuscript. We were also extremely gratified at the reviewer’s comments regarding the high quality of the writing and the science of our manuscript.

1. The author should choose between “algae” or “alga”. Unfortunately, alga is singular and algae is plural, so both are correct within the correct context. We have ensured we have used this use of these terms correctly in each case.

2. Page 7 line 262, should be C10H13OCl2Br2. Done.

3. Instrument details for optical rotation measurement should be included. Done.

Round  2

Reviewer 2 Report

The figure S23 added in Supplementary Material, to clarify the way in which the authors made the ECD calculation, does not satisfy my requests at all. Both the intensity, and the frequency scaling of the calculated spectrum, have no scientific base (if not that of better reproducing the experimental data). Moreover, nothing is said, again, about the conformational space search for molecule 7, neither about the UV spectrum calculation.

Author Response

Many thanks to the anonymous reviewer for their comments. We appreciate the fact that they are trying to make sure our science is robust.  As noted in the updated text, references have been provided where both wavelength and intensity scaling have been used previously to provide literature precedent for our approach. Also, the details of the relaxed scan which was used in place of conformational analysis have been incorporated in the manuscript. We hope this now clarifies the reviewers’ concerns. For ECD and UV calculations, the details have already been provided in that Gaussian was employed using a single point energy calculation with 24 excited states. These details are not given in the manuscript as these are technical details of the software.